# ADAR-mediated regulation of PQM-1 expression in neurons impacts gene expression throughout *C. elegans* and regulates survival from hypoxia

Ananya Mahapatra[1], Alfa Dhakal[2], Aika Noguchi[3], Pranathi Vadlamani[4], Heather A. Hundley [3]*

1 Genome, Cell and Developmental Biology Graduate Program, Indiana University, Bloomington, Indiana, United States of America, 2 Cell, Molecular and Cancer Biology Graduate Program, Indiana University School of Medicine–Bloomington, Bloomington, Indiana, United States of America, 3 Department of Biology, Indiana University, Bloomington, Indiana, United States of America, 4 Medical Sciences Program, Indiana University School of Medicine–Bloomington, Bloomington, Indiana, United States of America

* hahundle@indiana.edu

**Data Availability Statement:** All relevant data are within the paper and its Supporting Information files.

## Abstract

The ability to alter gene expression programs in response to changes in environmental conditions is central to the ability of an organism to thrive. For most organisms, the nervous system serves as the master regulator in communicating information about the animal's surroundings to other tissues. The information relay centers on signaling pathways that cue transcription factors in a given cell type to execute a specific gene expression program, but also provide a means to signal between tissues. The transcription factor PQM-1 is an important mediator of the insulin signaling pathway contributing to longevity and the stress response as well as impacting survival from hypoxia. Herein, we reveal a novel mechanism for regulating PQM-1 expression specifically in neural cells of larval animals. Our studies reveal that the RNA-binding protein (RBP), ADR-1, binds to *pqm-1* mRNA in neural cells. This binding is regulated by the presence of a second RBP, ADR-2, which when absent leads to reduced expression of both *pqm-1* and downstream PQM-1 activated genes. Interestingly, we find that neural *pqm-1* expression is sufficient to impact gene expression throughout the animal and affect survival from hypoxia, phenotypes that we also observe in *adr* mutant animals. Together, these studies reveal an important posttranscriptional gene regulatory mechanism in *Caenorhabditis elegans* that allows the nervous system to sense and respond to environmental conditions to promote organismal survival from hypoxia.

## Introduction

Aerobic heterotrophs need to obtain nutrition and oxygen from the environment, the prolonged absence of which can lead to undesirable consequences including death. However, fluctuations in oxygen and nutrient availability are common in nature and during development;

**Funding:** This work was supported by the National Science Foundation (Award 191750 to HAH), National Institute of Health/National Institute of General Medical Sciences (R01 GM130759 to HAH) and the John R. and Wendy L. Kindig Fellowship (to AM). Some strains were provided by the CGC, which is funded by NIH Office of Research Infrastructure Programs (P40 OD010440). The funders had no role in study design, data collection and analysis, decision to publish, or preparation of the manuscript.

**Competing interests:** The authors have declared that no competing interests exist.

**Abbreviations:** cDNA, complementary DNA; ChIP, chromatin immunoprecipitation; ILP, insulin-like peptide; RBP, RNA-binding protein; RIP, RNA immunoprecipitation; RNAi, RNA interference; UTR, untranslated region.

thus, organisms must have a means to both sense the environment and respond. At the most extreme, animals can effectively halt developmental and cellular programs resulting in a transient quiescent state [1]. For example, in the model organism *Caenorhabditis elegans* (*C. elegans*), the absence of oxygen can lead to a state of "suspended animation" [2], while first larval stage (L1) animals hatched in the absence of food enter a state of halted development commonly referred to as "L1 arrest" [3,4].

An animal's ability to enact these responses relies on the presence of a nervous system that can translate environmental information into physiological responses [5]. Neural gene expression programs are critical for organismal survival to many stresses. However, the nervous system must also communicate information about the environment to other tissues to promote diverse outputs, including behaviors and metabolic changes needed for organismal survival [6–8]. This trans-tissue communication is difficult to study in humans, but, altered brain–gut communication has been implicated in both oncogenesis and neurodegenerative diseases [9,10]. In contrast, studies in model organisms have been instrumental in demonstrating that the nervous system signals to the peripheral tissues to promote survival and longevity in response to stress [11]. Furthermore, recent data indicates that not only do some peripheral tissues receive the stress signals from the nervous system, but tissues like the intestine can also serve as an important regulatory organ, sending signals back to the nervous system to promote health and longevity [12].

The molecular players underlying the response to environmental conditions are conserved signaling pathways. The major signaling pathways that respond to nutrients include the AMP-activated protein kinase and Target of Rapamycin pathways, while the hypoxia-inducible factor 1 pathway responds to low oxygen (hypoxia) [13,14]. In addition, there is cross-talk between the insulin signaling pathway and all of these pathways [14–16], thereby making insulin signaling a key determinant of how animals respond to diet and environmental fluctuations in oxygen levels. In *C. elegans*, the insulin signaling pathway has only 1 known receptor, DAF-2, which is homologous to both the mammalian insulin receptor and insulin-like growth factor 1 receptor [17]. Despite having only 1 receptor, there are over 40 insulin-like peptides (ILPs) encoded in the *C. elegans* genome [18]. Binding to these agonists and antagonists can influence the ability of DAF-2 to signal to downstream kinases that in turn regulate at least 2 transcription factors, the well-established FOXO homolog DAF-16 and/or the more recently identified zinc finger protein PQM-1 [19,20].

Genetic dissection of the insulin signaling pathway, particularly the study of temperature-sensitive loss of function mutations of *daf-2*, has identified a central role for this pathway in regulating development, longevity, metabolism, and reproduction [19,21,22]. In addition to genomic mutations, which impact signaling throughout the animal, it is well established that the nervous system is the critical site of action for insulin signaling to regulate diverse aspects of *C. elegans* physiology [23]. For example, neuronal-specific expression of the insulin receptor rescues both the long-lived phenotype observed in adult animals with altered DAF-2 function as well as the formation of dauer larva, a developmentally arrested life stage that is induced by overcrowding and starvation in wild-type *C. elegans* but constitutively occurs in *daf-2* mutant larval animals [24,25]. Recently, a novel regulatory mechanism for altering insulin signaling via alternative splicing of *daf-2* in neurons was identified [26]. The resulting DAF-2B protein retains the extracellular domain but lacks the intracellular domains to mediate downstream signaling, which allows the DAF-2B protein to bind ILPs and influence insulin signaling by competing with full-length DAF-2. Consistent with this, the presence of DAF-2B influences dauer entry and recovery as well as lifespan, further supporting the idea that nervous system-specific regulation of the insulin signaling pathway is important.

Previous studies from our lab identified adenosine (A) to inosine (I) RNA-editing sites in *daf-2* mRNA isolated from neural cells of L1-arrested animals [27]. Due to differences in base-pairing properties of adenosine and inosine, A-to-I editing events can impact gene expression depending on the region of RNA in which the editing event occurs [28,29]. For example, A-to-I editing within coding sequences of genes can alter the protein encoded by the gene and editing within 3′ untranslated regions (UTRs) can alter small RNA binding [30]. For the *daf-2* transcript, the A-to-I editing sites identified are located within an intronic sequence, which could potentially impact the production of *daf-2* splice isoforms. To begin to understand if RNA editing influences DAF-2 function, we examined gene expression changes that occur in neural cells in the absence of the enzyme that is responsible for catalyzing the hydrolytic deamination of adenosine to inosine, ADR-2. ADR-2 is a member of the adenosine deaminase that act on RNA (ADAR) family, and the *C. elegans* genome encodes 2 ADAR family members ADR-1 and ADR-2 [31]. ADR-2 is the sole A-to-I editing enzyme in *C. elegans* [32,33], as ADR-1 lacks essential amino acids required to perform deamination [32]. However, as ADARs are RNA-binding proteins (RBPs) that can also regulate gene expression through binding RNA [34], both ADR-1 and ADR-2 may play roles in editing-independent gene regulation in *C. elegans*. Interestingly, the long-lived phenotype of animals lacking *daf-2* is also observed in *adr-2(-)* mutants [24,35]. Together, these data suggested the possibility of *C. elegans* ADARs impacting insulin signaling. While a role for ADARs in regulating insulin signaling is relatively unexplored, recent work in β cells indicated that the pathophysiological environment of type 1 diabetes patients influences RNA editing [36]. In this work, we sought to determine how ADARs can affect the insulin signaling pathway, particularly in the nervous system of L1-arrested *C. elegans*.

## Results

### Decreased expression of genes regulated by insulin signaling upon loss of *adr-2*

As a first step towards addressing whether ADR-2 regulates insulin signaling, the transcriptomes of wild type and *adr-2*-deficient animals were compared. As editing of *daf-2* was observed in neural cells isolated from synchronized L1 animals [27], differential gene expression was analyzed in RNA isolated from these same types of biological samples. Using datasets from previously performed RNA-sequencing (RNA-seq) of 3 biological replicates of wild type and *adr-2(-)* neural cells from synchronized L1 animals [37], differential gene expression analysis identified 697 genes significantly altered in neural cells from *adr-2(-)* animals ($p$ value $< 0.05$ and log$_2$fold change $> |0.5|$), with nearly 3 times as many down-regulated genes (501) as up-regulated genes (196) (Fig 1A, S1 Table). These misregulated genes were subjected to gene set enrichment analysis using a *C. elegans* specific software, WormCat [38]. The analysis for genes up-regulated in neural cells from *adr-2(-)* animals revealed only 1 significantly enriched gene set, extracellular material (S1 Fig). The down-regulated genes were enriched for 4 gene sets: stress response, proteolysis, metabolism, and lysosome (S1 Fig). As DAF-2-mediated signaling regulates both stress response and metabolism [19], this suggests that loss of *adr-2* might result in altered insulin signaling in neural cells. However, it should be noted that these broad categories of gene sets can be influenced by many factors, and we did not observe significant changes in *daf-2* mRNA expression in *adr-2(-)* neural cells (S1 Table).

To independently examine the expression of genes regulated by DAF-2, qPCR was performed for 3 known downstream targets in 3 independent biological replicates of neural cells isolated from wild-type and *adr-2(-)* animals. Consistent with the RNA-seq dataset, all 3 genes (*dod-17*, *dod-19*, and *dod-24*) examined were significantly down-regulated in *adr-2(-)* neural

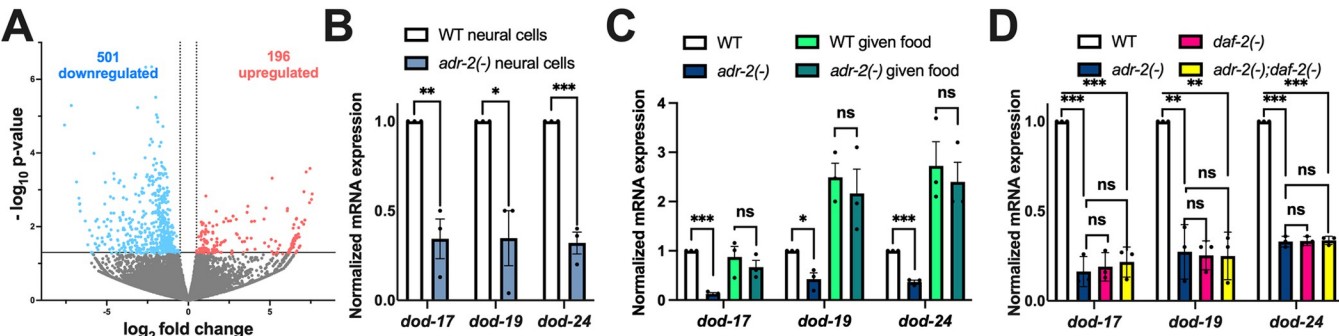

**Fig 1. L1 animals lacking *adr-2* have decreased expression of genes regulated by insulin signaling.** (A) Volcano plot depicting gene expression in *adr-2(-)* neural cells compared to WT neural cells. Dots represent individual genes that are up-regulated (red; 186, $p < 0.05$, log2fold > 0.5), down-regulated (blue; 502, $p < 0.05$, log2fold < -0.5), or not significantly different (gray, $p \geq 0.05$) between 3 biological replicates of WT and *adr-2(-)* neural cells. (B–D) Expression of the indicated genes was determined relative to expression of the housekeeping gene *gpd-3*. Values were then normalized to WT neural cells (B) or WT L1 animals hatched in the absence of food (C, D) and the mean of 3 biological replicates was plotted. Error bars represent SEM. Statistical significance was calculated using multiple unpaired *t* tests followed by Holm–Šídák multiple comparisons correction. *$p < 0.05$, **$p < 0.005$, ***$p < 0.0005$, ns indicates not significant ($p > 0.05$). For D, the indicated genotypes are of strains HAH30, HAH31, HAH32, and HAH33. All individual data and statistics are included in S1 Data under Supporting information. SEM, standard error of the mean; WT, wild type.

cells compared to wild-type neural cells (Fig 1B). These results suggest that, within neural cells, there is decreased expression of genes regulated by DAF-2 upon loss of *adr-2*.

While this data suggests that loss of *adr-2* affects genes regulated by insulin signaling in the nervous system, the nervous system is also the master regulator that coordinates gene regulation between tissues [39]. Studies have demonstrated that DAF-2 function in the nervous system can affect phenotypes such as organismal lifespan by signaling to other tissues [25]. This raised the question of whether the decreased expression of DAF-2-regulated genes upon *adr-2* loss could be observed in RNA isolated from whole L1 animals. Furthermore, the neural cells were isolated from L1 animals that were synchronized by hatching in the absence of food, and nutrient levels impact insulin signaling [40]. Hence, gene expression was examined in RNA isolated from 3 independent biological replicates of synchronized wild-type and *adr-2(-)* L1 animals as well as a subset of these hatched L1 animals that were exposed to bacterial food for 6 h. Similar to neural cells, all 3 genes (*dod-17*, *dod-19*, and *dod-24*) exhibited significantly decreased expression in hatched L1 animals lacking *adr-2* compared to wild-type animals (Fig 1C). This suggests that the impacts of loss of *adr-2* on altered neural gene expression (Fig 1A and 1B) may lead to cell non-autonomous effects and/or that *adr-2* regulates genes downstream of insulin signaling in several tissues. In contrast to the hatched L1 animals, there was no significant difference in expression of the 3 genes (*dod-17*, *dod-19*, and *dod-24*) between wild-type and *adr-2(-)* animals after feeding for 6 h (Fig 1C).

Together, these data suggest that upon loss of *adr-2*, reduced expression of genes regulated by insulin signaling occurs in whole animals and is abrogated by the presence of food. Since the observed decreased gene expression was specific to starved *adr-2(-)* animals and insulin signaling is known to impact starvation responses in *C. elegans*, these data suggested that *adr-2* regulates *dod-17*, *dod-19*, and *dod-24* through the insulin signaling pathway. To directly test this possibility, genetic mutants of *adr-2* and *daf-2* [41] were combined and gene expression was examined. As observed consistently in this study, compared to wild-type animals, *dod-17*, *dod-19*, and *dod-24* were all significantly reduced in *adr-2(-)* animals (Fig 1D). Consistent with previous studies [42], *dod-17*, *dod-19*, and *dod-24* all exhibited significantly decreased expression in *daf-2(-)* animals (Fig 1D). Compared to wild-type animals, there was a significant

reduction in mRNA expression of all 3 insulin signaling regulated genes in the *adr-2(-);daf-2 (-)* double mutants (Fig 1D). Additionally, the expression profile of *dod-17*, *dod-19*, and *dod-24* was similar between the *adr-2(-)* and *daf-2(-)* single mutants and the *adr-2(-);daf-2(-)* double mutants (Fig 1D). Together, these experiments indicate that lack of *adr-2* leads to decreased expression of insulin signaling regulated genes and that both ADR-2 and DAF-2 function in the same pathway to regulate expression of the *dod* genes.

## Regulation of insulin signaling by ADR-2 is cell non-autonomous and editing-independent

In addition to being expressed in the nervous system, genes regulated by insulin signaling are also highly expressed in the intestine [43,44]. As the RNA isolated from L1 animals hatched in the absence of food shows decreased gene expression similar to neural cells, this suggests that loss of *adr-2* can potentially result in altered insulin-signaling in both neural and intestinal cells. However, there is extensive communication between neural cells and the intestine; thus, it is possible that loss of *adr-2* in neural cells is sufficient to result in altered insulin signaling in the intestine. To test this possibility, gene expression was monitored in animals that express ADR-2 only in the nervous system [37]. Briefly, these transgenic animals were generated by injecting *adr-2(-)* animals with a plasmid construct in which a pan-neural promoter *rab-3* drives the expression of *adr-2* along with a co-injection marker expressing GFP. A similar strain was previously generated by our lab and shown to result in ADR-2 activity in the nervous system [37]. Since transgenes do not exhibit 100% inheritance in *C. elegans*, the transgenic animals of interest were sorted for GFP expression using the COPAS Select large particle sorter. Optimized COPAS conditions in terms of extinction and time of flight were used to create a gated window for specifically sorting L1 animals. RNA was isolated from the sorted animals and qPCR was performed to assess gene expression. It was observed that compared to *adr-2* lacking animals, animals expressing ADR-2 solely in the nervous system significantly rescued expression of genes regulated by insulin signaling (Fig 2A). This data suggests that the presence of ADR-2 in the nervous system regulates insulin signaling throughout the animals.

To further validate that neural ADR-2 regulates insulin signaling regulated genes throughout the animal, confocal microscopy was performed to monitor expression of one of these genes, *dod-24*, upon loss of *adr-2* in whole animals. Synchronized L1 animals expressing GFP driven by the *dod-24* promoter [43] were analyzed. In wild-type animals, transcription from the *dod-24* promoter was observed in neural as well as intestinal cells as expected [43] (Fig 2B). Upon loss of *adr-2*, decreased GFP expression was observed throughout the animal (Fig 2B), which is consistent with the qPCR analysis of *dod-24* expression (Fig 1C). Furthermore, in animals expressing ADR-2 solely in the nervous system, GFP expression was similar to wild-type animals (Fig 2B). These results demonstrate that the presence of ADR-2 in the nervous system cell non-autonomously impacts gene expression.

To begin to dissect the molecular function of ADR-2 in the nervous system that contributed to the altered gene regulation, expression of *dod-17*, *dod-19*, and *dod-24* was examined in animals expressing an ADR-2 mutant (ADR-2 G184R) that can bind RNA, but lacks the ability to edit [27]. As observed consistently in this study, hatched L1 *adr-2(-)* animals had significantly decreased expression of insulin signaling regulated genes compared to wild-type animals (Fig 2C). In contrast, gene expression in the ADR-2 G184R animals was similar to wild-type animals (Fig 2C). Together, these results indicate that, while the presence of ADR-2 in the nervous system is critical for proper *dod-17*, *dod-19*, and *dod-24* expression throughout the animal, the editing function of ADR-2 is not required for this gene regulatory function.

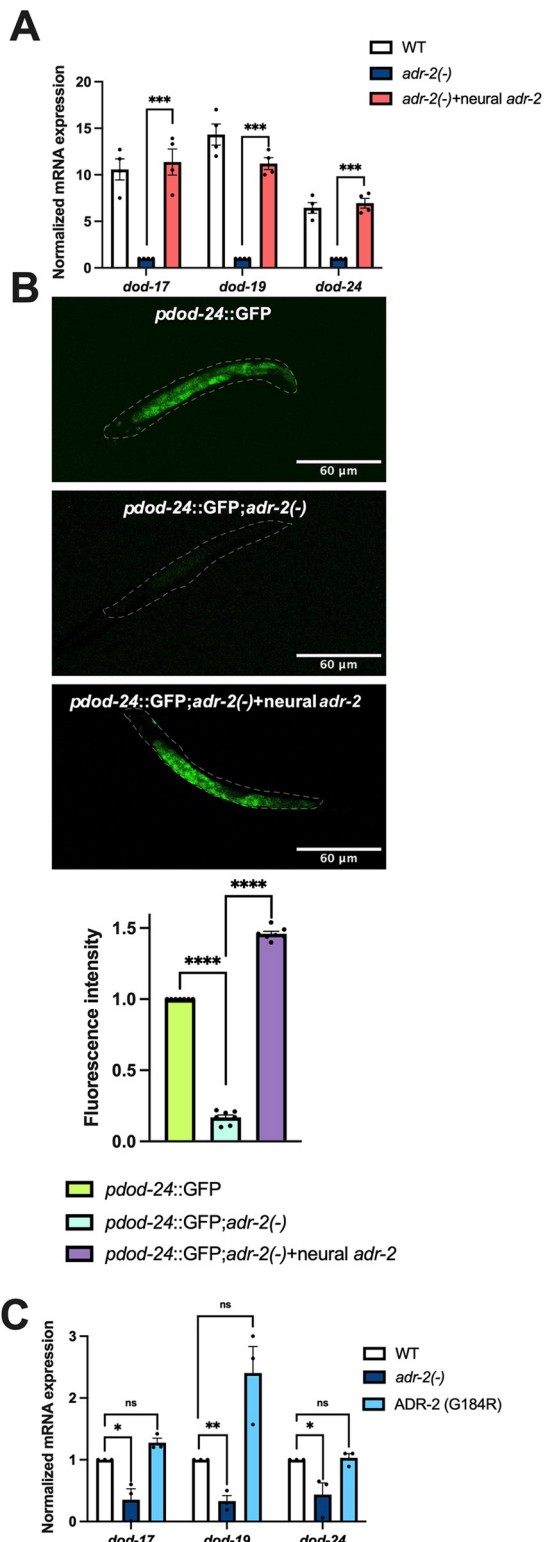

**Fig 2. Neural ADR-2 regulates insulin signaling cell non-autonomously in an editing-independent manner.** (A, C) Gene expression of L1-arrested animals measured by qPCR. Expression of the indicated genes was determined relative to expression of the housekeeping gene *gpd-3*. Values were then normalized to WT and the mean of 3 (C) or 4 (A) biological replicates was plotted. Error bars represent SEM. Statistical significance was calculated using multiple unpaired *t* tests followed by Holm–Šídák multiple comparisons correction. ***$p < 0.0005$, ns indicates not significant

($p > 0.05$). For A, the indicated genotypes are of strains HAH23, HAH40, and HAH41. For C, the indicated genotypes are of strains N2, BB20, and HAH22. (B) A representative image of 1 L1 animal of the indicated genotypes. The dashed line represents the outline of the whole worm. For all the strains, the images are representative of 7 samples imaged in 2 biological replicates. The bar graphs below the images represent the summary of fluorescence intensity quantification using FIJI software for all the animals imaged. Data from 7 animals are plotted where each dot represents 1 animal. The values were normalized to *pdod-24*::GFP. Statistical significance was determined using an ordinary one-way ANOVA test. ****$p < 0.0001$. All individual data and statistics are included in S2 Data. SEM, standard error of the mean; WT, wild type.

## Neural *pqm-1* levels impact downstream gene expression throughout the animal

As the ADR-2 editing function was not required for altered *dod-17*, *dod-19*, and *dod-24* expression, editing of *daf-2* is unlikely to be causing the decreased expression of downstream insulin signaling regulated genes in neural cells and animals lacking *adr-2*. As DAF-2 is at the top of the insulin-signaling regulatory cascade and transcriptional output is mediated by at least 2 different transcription factors, DAF-16 and PQM-1, we sought to examine whether all DAF-2 regulated genes were equally affected by loss of *adr-2*. The up- and down-regulated genes in the *adr-2(-)* neural RNA-seq dataset were individually overlapped with either DAF-16 activated or PQM-1 activated genes from a published dataset [20] (S2 Fig). The number of PQM-1 activated genes that were down-regulated in *adr-2(-)* neural cells (156) was nearly 3 times the number that would be expected by random chance (53) (S2 Fig) and *dod-17*, *dod-19*, and *dod-24* are all genes activated by PQM-1 [20]. The number of overlapping PQM-1 activated genes that were up-regulated in *adr-2(-)* neural cells (11) was lesser than that obtained due to random chance (20) (S2 Fig). Further, the number of overlapping DAF-16 activated genes either up- (23) or down-regulated (61) in *adr-2(-)* neural cells was very close to what would be expected from random chance (S2 Fig). These results suggest that loss of *adr-2* does not impact all genes downstream of the insulin signaling pathway, but instead, leads to specific down-regulation of PQM-1-activated genes.

The above data raised the question of whether loss of *adr-2* directly impacts *pqm-1* expression. To address this question, *pqm-1* expression was monitored in both neural cells and L1 animals using qPCR. In the neural cells from *adr-2(-)* animals, there was a significant decrease in *pqm-1* expression compared to neural cells isolated from wild-type animals (Fig 3A). Consistent with this finding, our neural RNA-seq datasets also revealed that loss of *adr-2* indeed resulted in significantly decreased neural expression of *pqm-1* (S1 Table). In contrast to neural cells, *pqm-1* expression was not significantly altered in RNA isolated from synchronized L1 animals lacking *adr-2* (Fig 3B). Together, this data indicates that the lack of *adr-2* impacts *pqm-1* expression in a tissue-specific manner. The data also suggests that decreased neural expression of *pqm-1* upon loss of *adr-2* could impact gene expression throughout the animal.

To directly address whether the decreased expression of PQM-1-activated genes in *adr-2(-)* animals is due to decreased *pqm-1* expression, gene expression was assessed in animals lacking *pqm-1(-)* and *adr-2(-);pqm-1(-)* animals. Briefly, *pqm-1(ok485)* animals [45] were obtained and backcrossed to wild-type animals before crossing to *adr-2(-)* animals. RNA was then isolated from wild type, *adr-2(-)*, *pqm-1(-)*, and *adr-2(-);pqm-1(-)* animals obtained from the genetic cross. As expected, compared to wild-type animals, *adr-2(-)* animals showed significantly decreased expression of *dod-17*, *dod-19*, and *dod-24* (Fig 3C). Similarly, loss of *pqm-1* resulted in significantly decreased expression of *dod-17*, *dod-19*, and *dod-24* compared to wild-type animals (Fig 3C). These PQM-1 activated genes also showed a significant reduction in expression in *adr-2(-);pqm-1(-)* animals (Fig 3C), which was similar to the expression levels observed in the individual *adr-2(-)* and *pqm-1(-)* mutant animals (Fig 3C). These results suggest that the

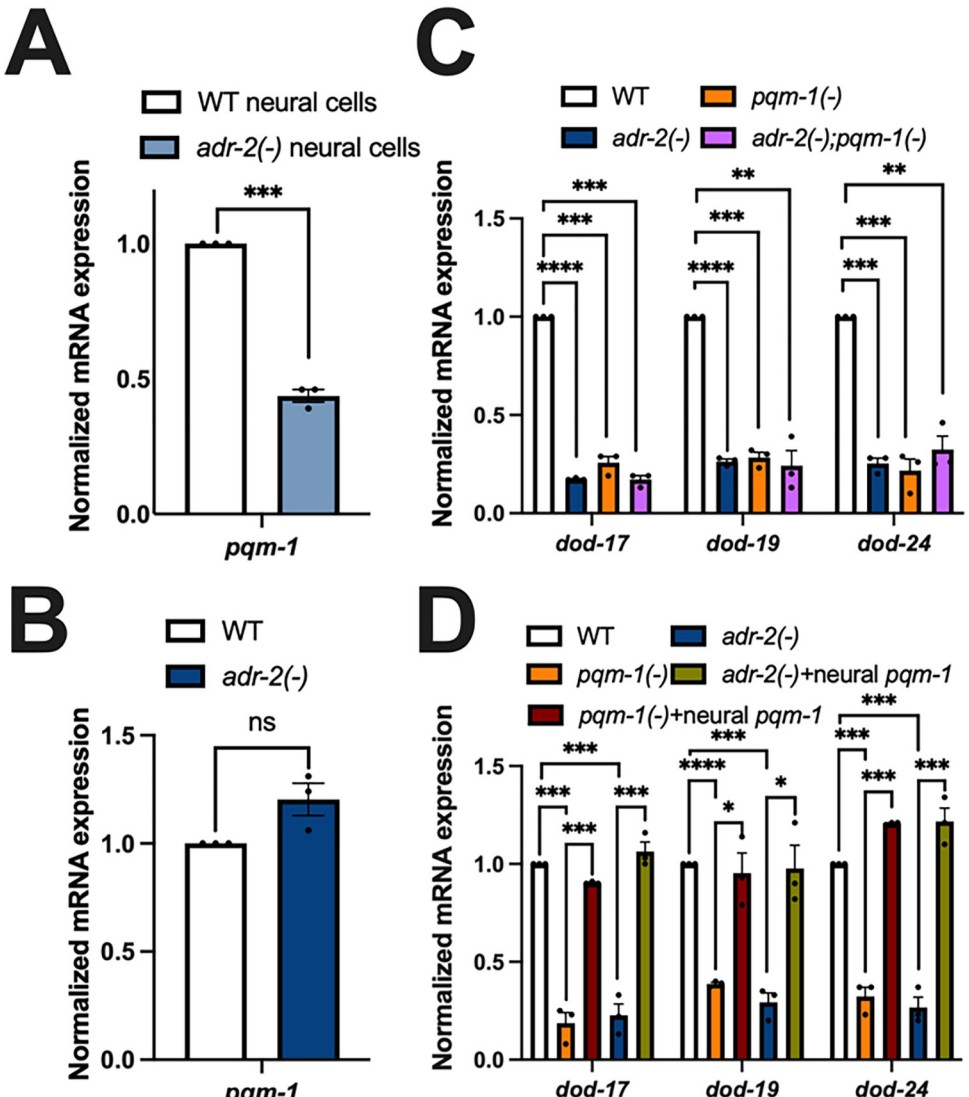

**Fig 3. Neural PQM-1 activity is sufficient to rescue expression of PQM-1 activated genes in *adr-2(-)* animals.** (A–D) Gene expression of (A) neural cells and (B–D) L1-arrested animals measured by qPCR. Expression of indicated genes was determined relative to expression of the housekeeping gene gpd-3. Values were then normalized to WT and the mean of 3 biological replicates was plotted. Error bars represent SEM. Statistical significance was calculated using multiple unpaired *t* tests followed by Holm–Šídák multiple comparisons correction. $**p < 0.005$, $***p < 0.0005$, $****p < 0.000001$, ns indicates not significant ($p > 0.05$). For A and B, the indicated genotypes are of strains HAH45 and HAH46. For C, the indicated genotypes are of strains HAH35, HAH37, HAH38, and HAH39. For D, the indicated genotypes are of strains HAH24, HAH41, HAH42, HAH43, and HAH44. All individual data and statistics are included in S3 Data under Supporting information. SEM, standard error of the mean; WT, wild type.

decreased expression of *dod-17*, *dod-19*, and *dod-24* upon loss of *adr-2* occurs via altered PQM-1 function.

As our results clearly indicate that loss of *adr-2* leads to down-regulation of *pqm-1* specifically in the nervous system, but decreased expression of PQM-1 activated genes throughout L1 animals, we sought to test whether expressing *pqm-1* only within the nervous system of *adr-2 (-)* animals could restore *dod-17*, *dod-19*, and *dod-24* gene expression in L1 animals. Transgenic animals were generated by injecting a plasmid in which *pqm-1* expression is driven by the neuronal *rab-3* promoter. As a control, this plasmid was first injected into *pqm-1(-)*

animals to observe gene expression changes in *pqm-1(-)* animals. The resulting transgenic animals were crossed with *adr-2(-)* animals and genotyped for either *pqm-1(-)* or *adr-2(-)* animals that specifically express *pqm-1* in the nervous system. Compared to wild-type animals and consistent with other results in this study, there was significantly decreased expression of *dod-17*, *dod-19*, and *dod-24* in *pqm-1(-)* animals (Fig 3D). However, *pqm-1(-)* animals expressing *pqm-1* only in the nervous system had significantly increased expression of *dod-17*, *dod-19*, and *dod-24*, which rescued the gene expression to near wild-type levels (Fig 3D). As observed throughout this study, *adr-2(-)* animals exhibited significantly decreased expression of PQM-1 activated genes compared to wild-type animals (Fig 3D). However, *adr-2(-)* animals carrying the neural *pqm-1* transgene exhibited significantly increased expression of *dod-17*, *dod-19*, and *dod-24* compared to animals lacking *adr-2* (Fig 3D). Together, these data indicate that lack of *adr-2* leads to global down-regulation of *dod-17*, *dod-19*, and *dod-24* via decreased expression of the PQM-1 transcription factor in the nervous system.

## In the absence of *adr-2*, ADR-1 binds to *pqm-1* mRNA and results in decreased expression of PQM-1 activated genes

The above data suggest that loss of *adr-2* results in decreased *pqm-1* expression in the nervous system, which leads to decreased expression of PQM-1 activated genes throughout the animal. Since the editing function of ADR-2 is not required for the down-regulation of these PQM-1 activated genes (Fig 2C), we sought to test what other function of ADR-2 was critical for regulating *pqm-1* expression. ADR-2 directly interacts with ADR-1, a deaminase-deficient member of the ADAR family present in *C. elegans* [46]. The physical interaction between ADR-1 and ADR-2 can both promote ADR-2 binding to RNA [46], as well as influence the RNAs that ADR-1 binds [35]. However, the biological impacts of this latter function are relatively unknown. To assess if PQM-1 activation of gene expression is altered upon loss of *adr-1*, RNA was isolated from hatched wild type, *adr-1(-)*, *adr-2(-)*, and *adr-1(-);adr-2(-)* L1 animals, and qPCR was performed to measure gene expression of *dod-17*, *dod-19*, and *dod-24* in these animals. Consistent with the data obtained in this study, *adr-2(-)* animals had decreased expression of the PQM-1 activated genes compared to wild-type animals (Fig 4A). In contrast, expression levels of *dod-17*, *dod-19*, and *dod-24* were not significantly different between wild-type and *adr-1(-)* animals (Fig 4A), suggesting that loss of ADR-1 function did not affect PQM-1-mediated gene regulation in wild-type animals. Interestingly, loss of *adr-1* in animals lacking *adr-2* significantly increased expression of PQM-1 activated genes compared to animals lacking only *adr-2* (Fig 4A). These results suggest that ADR-1 has a unique function specifically in the absence of *adr-2*, which results in decreased expression of PQM-1 activated genes.

So far, the data suggests that upon loss of *adr-2*, decreased *pqm-1* expression in neural cells leads to global down-regulation of PQM-1 activated genes and loss of *adr-1* can rescue these downstream gene expression changes. As ADR-1 is an RBP, we sought to determine whether ADR-1 directly binds *pqm-1* mRNA specifically in the nervous system, and whether that binding is influenced by the presence or absence of *adr-2*. To examine binding of ADR-1 to *pqm-1* in the L1 nervous system, an RNA immunoprecipitation (RIP) assay was performed with animals that express ADR-1 specifically in the nervous system. To generate these animals *adr-1(-)* animals were injected with a construct in which the neuronal *rab-3* promoter drives expression of an N-terminally 3X FLAG *adr-1* genomic sequence. A similar epitope tagged construct under the control of the *adr-1* endogenous promoter was previously demonstrated to produce functional ADR-1 protein [33] and has been used in previous studies to examine ADR-1 RNA binding [33,35]. Hatched L1 animals were subjected to UV crosslinking to stabilize RNA–

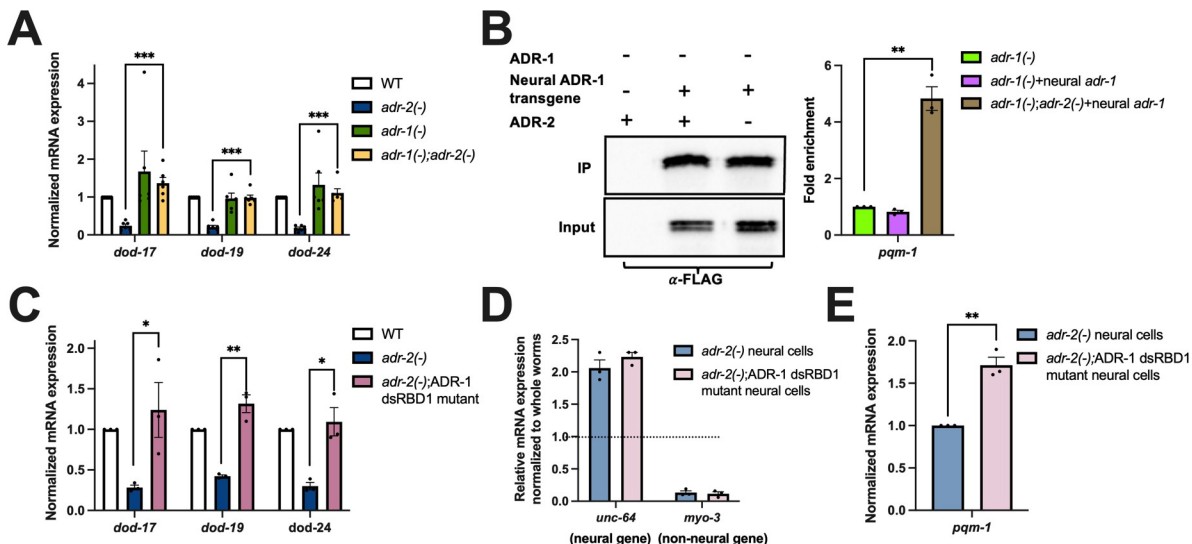

**Fig 4. Neural ADR-1 binding of *pqm-1* affects expression of PQM-1 activated genes.** (A, C, E) Gene expression of (A, C) L1-arrested animals and (E) neural cells measured by qPCR. Expression of the indicated genes was determined relative to expression of the housekeeping gene *gpd-3*. Values were then normalized to WT L1 animals (A, C) or *adr-2(-)* neural cells (E) and the mean of (A) six (C, E) 3 biological replicates was plotted. Error bars represent SEM. Statistical significance was calculated using multiple unpaired *t* tests followed by Holm–Šídák multiple comparisons correction. *$p < 0.05$, **$p < 0.005$, ***$p < 0.0005$. For A, the indicated genotypes are of strains N2, BB19, BB20, and BB21. For C, the indicated genotypes are of strains HAH48, HAH49, and HAH50. (B) Western blot depicting immunoprecipitation of neural ADR-1 from the indicated strains. Bar graph represents the fold enrichment of *pqm-1* cDNA in the IP samples relative to the amount of *pqm-1* cDNA in the input lysate for each strain. The IP/input values are obtained for each strain and then normalized to the IP/input value for the negative control (*adr-1*(-)). The mean of 3 biological replicates was plotted. Error bars represent SEM. Statistical significance was calculated by multiple unpaired *t* tests followed by Holm–Šídák multiple comparisons correction. **$p < 0.005$. (D) Gene expression of neural *unc-64* and nonneural *myo-3* measured by qPCR. Expression was determined relative to *gpd-3* and values were normalized to nonneural cells. All individual data and statistics are included in S4 Data under Supporting information. cDNA, complementary DNA; SEM, standard error of the mean; WT, wild type.

protein interactions prior to generation of protein lysates. ADR-1 and associated RNAs were immunoprecipitated using magnetic FLAG beads and then a portion of the immunoprecipitation was taken for western analysis while the remaining was treated with Proteinase K to release bound RNA. To account for the nonspecific binding of RNA to the magnetic FLAG beads, immunoprecipitation and RNA isolation was also performed from *adr-1(-)* animals. From immunoblotting analysis, ADR-1 was found to be efficiently immunoprecipitated from animals expressing ADR-1 in the nervous system both in the presence and absence of *adr-2*, but not from lysates of the negative control *adr-1(-)* animals (Fig 4B). On assessing *pqm-1* mRNA in the assay, compared to the negative control, there was no enrichment for *pqm-1* mRNA in IPs from neural ADR-1-expressing animals that expressed wild-type ADR-2 (Fig 4B). However, a 5-fold enrichment of *pqm-1* mRNA was observed in the neural ADR-1 RIP in the absence of *adr-2* (Fig 4B). This data suggests that *in vivo*, ADR-1 binds *pqm-1* in the nervous system, but only in the absence of *adr-2*.

To further examine whether the RNA-binding function of ADR-1 is contributing to the decreased expression of PQM-1 activated genes observed in *adr-2(-)* animals, *dod-17*, *dod-19*, and *dod-24* expression was monitored in *adr-2* lacking animals that also have abolished ADR-1 binding. These mutant animals have 3 mutations within the conserved KKxxK motif (where K is lysine and x is any amino acid) of the first dsRNA binding domain (dsRBD1) of ADR-1 (K223E, K224A, K227A), which was previously shown to disrupt the ability of ADR-1 to bind RNA *in vivo* [46]. The mutation was introduced in wild-type animals with an integrated 3X FLAG tag at the *adr-1* locus via CRISPR by using a guide RNA targeted to the *adr-1* locus and

an HDR template containing the desired mutations. These animals were then crossed to *adr-2 (-)* animals to generate *adr-2(-)* animals that also lack the ADR-1 binding function. RNA was isolated from these animals as well as wild-type and *adr-2(-)* animals and qPCR was performed. Compared to *adr-2(-)* animals, *adr-2(-);* ADR-1 dsRBD1 mutants had significantly increased expression of *dod-17*, *dod-19*, and *dod-24* (Fig 4C).

These data suggest that in the absence of *adr-2*, ADR-1 binds *pqm-1* in the nervous system and leads to decreased *pqm-1* expression, which in turn impacts expression of PQM-1 activated genes throughout the animal. To gain further insight into this model, *pqm-1* expression was assessed in *adr-2(-)* mutants lacking the ADR-1 binding function. First, successful neural cell isolation was validated by measuring the expression of *unc-64* (neural gene) and *myo-3* (nonneural gene). Compared to nonneural cells, there was an enrichment for *unc-64* but no enrichment for *myo-3* in the neural cells (Fig 4D), suggesting high purity of neural cells isolated. Compared to neural cells isolated from *adr-2(-)* animals, neural cells from *adr-2(-)* animals lacking ADR-1 binding exhibit a significant increase in *pqm-1* expression (Fig 4E). Furthermore, this increase is approximately 50% to 60%, which is similar to the decrease in *pqm-1* expression observed between neural cells isolated from wild-type and *adr-2(-)* animals (Fig 3A). Together, these results indicate that, in the absence of *adr-2*, ADR-1 binding to the *pqm-1* transcript in the nervous system causes decreased *pqm-1* expression and hence decreased expression of PQM-1 activated genes throughout the animal.

## PQM-1 functions in the nervous system to regulate hypoxia survival of L1-arrested animals

Previous studies have indicated that PQM-1 is a negative regulator of hypoxic survival in fourth stage larval (L4) animals [47]. As our data indicates that *adr-2(-)* animals have decreased *pqm-1* expression, we sought to determine if these animals also had altered survival to hypoxic exposure. Additionally, since the data so far indicate that ADARs regulate *pqm-1* expression specifically in the nervous system, whether neural PQM-1 specifically plays a role in survival to hypoxia was of interest.

To directly test these questions, hatched L1 animals were exposed to varying concentrations of cobalt chloride ($CoCl_2$), which serves as a hypoxia mimetic [48]. As we wanted to address the contribution of neural PQM-1 in regulating hypoxia, survival of the neural-specific *pqm-1* transgenic animals was examined. However, as the neural *pqm-1* strains were transgenic, to avoid any nonspecific effects of the coinjected transgenes, survival was compared in animals that all carry the *prab-3*::GFP transgene. Wild-type, *pqm-1(-)*, *pqm-1(-)* animals expressing *pqm-1* in the nervous system, *adr-2(-)* and *adr-2(-)* animals overexpressing *pqm-1* in the nervous system were used in the hypoxic survival experiment. After obtaining hatched L1 animals, 5,000 animals per strain were washed with NaCl and then exposed to varying concentrations of $CoCl_2$ (0 to 80 mM) for 2 h. After the $CoCl_2$ exposure, 3 technical replicates of 30 to 40 GFP positive L1 animals per strain were plated and incubated for 24 h at 20°C in the presence of food. To measure hypoxic survival, alive and dead animals were counted for all the strains and plotted for all concentrations of $CoCl_2$. At lower concentrations of $CoCl_2$ (2.5 mM and 5 mM), the number of alive L1s was similar across all worm strains (Fig 5A). However, at 10 mM and higher concentrations of $CoCl_2$, consistent with previous studies [49], there was a drastic reduction in the survival of wild-type animals (Fig 5A). With increasing concentrations of $CoCl_2$, the number of alive *pqm-1(-)* and *adr-2(-)* animals was significantly higher than wild-type animals (Figs 5A and S3A), suggesting an increased hypoxic survival of these animals. Strikingly, there was a sharp decline in the survival of *pqm-1(-)* animals expressing neural *pqm-1* compared to *pqm-1(-)* animals (Figs 5A and S3A). This data suggests that PQM-1

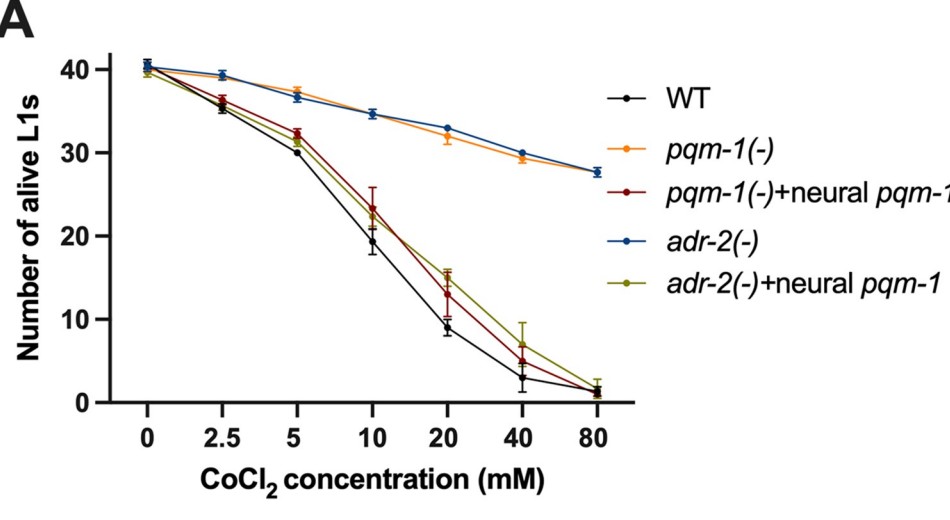

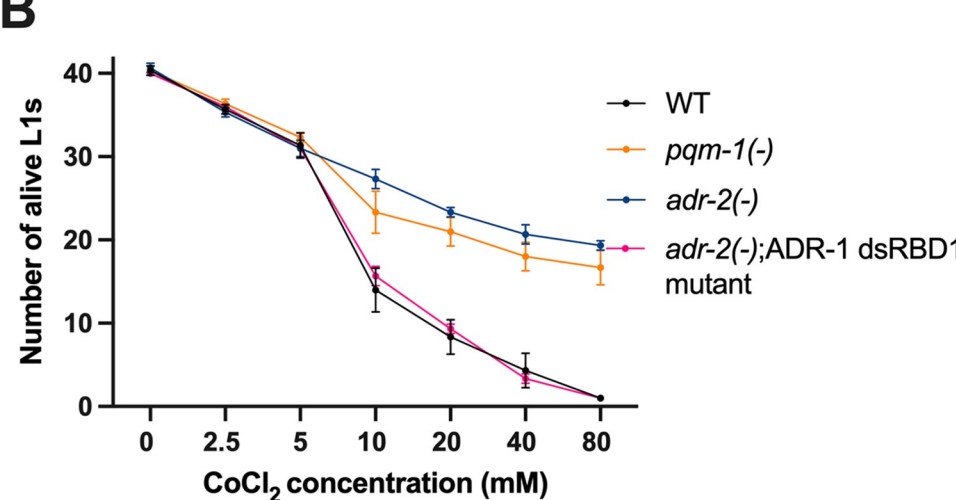

**Fig 5. Survival of hatched L1 animals after CoCl₂ exposure.** (A, B) Survival of transgenic (A) and non-transgenic (B) hatched L1 animals under hypoxic conditions induced by CoCl₂ exposure. Data plotted is average of 3 biological replicates. Error bars represent SEM. All individual data and statistics are included in S5 Data under Supporting information. SEM, standard error of the mean; WT, wild type.

function within the nervous system is sufficient to regulate hypoxic survival. Interestingly, a similar result was observed with *adr-2(-)* animals with transgenic expression of *pqm-1* in the nervous system (Figs 5A and S3A). Together, these data suggest that PQM-1 in the nervous system is a critical regulator of hypoxia survival in hatched L1 animals and that this function of PQM-1 impacts the survival of animals lacking *adr-2* in hypoxic environments.

As our molecular data suggest that ADR-1 binding to *pqm-1* mRNA in the absence of *adr-2* results in altered *pqm-1* expression, we sought to determine if loss of ADR-1 RNA binding could influence the survival of *adr-2(-)* animals to hypoxia. An independent set of hypoxia survival experiments were performed using wild type, *pqm-1(-)* and *adr-2(-)* animals along with the *adr-2(-)*;ADR-1 dsRBD1 mutant animals. Consistent with the transgenic animals assayed in our study, compared to wild-type animals, both *pqm-1(-)* and *adr-2(-)* animals showed a significantly increased hypoxic survival (Figs 5B and S3B). Compellingly, *adr-2(-)* animals lacking ADR-1 RNA-binding function showed hypoxic survival similar to wild-type animals (Fig 5B).

Together, these results suggest that neural PQM-1 is a key mediator of hypoxic survival and that binding of ADR-1 to *pqm-1* mRNA in the nervous system affects the animal's ability to survive hypoxic stress.

## Discussion

In these studies, we determined the tissue-specific contributions of ADAR proteins in regulating the insulin signaling pathway in *C. elegans*. Our data revealed unique ADR-1 RNA binding that occurs in the nervous system specifically in the absence of *adr-2*. Furthermore, our neural cell data indicate that the binding of ADR-1 in neural cells is sufficient to cause down-regulation of *pqm-1* transcript in the absence of *adr-2*. However, the molecular details of how ADR-1 binding leads to decreased *pqm-1* expression are an open question. Previous work from our lab has shown that ADR-1 binding to another transcript, *clec-41*, is important to promote neural gene expression [27]; however, that mechanism was editing and ADR-2 dependent. Editing-independent effects of ADARs on mRNA stability have been identified for human ADAR1 and ADAR2 [50,51]. However, unlike what we observe for *C. elegans* ADR-1 promoting decreased *pqm-1* expression, human ADAR1 and ADAR2 promoted mRNA stability by altering the interaction of bound transcripts with other RBPs, particularly HuR. It is possible that *C. elegans* ADR-1 also affects the binding of other RBPs to *pqm-1* and these factors are critical for stabilizing *pqm-1*. Alternatively, ADARs, including the *C. elegans* ADARs, are known to impact small RNA levels [52,53]. Perhaps the absence of *adr-2* results in altered small RNA expression and ADR-1 binding to *pqm-1* influences the ability of these small RNAs to bind *pqm-1* to promote decay. Future studies of factors that bind the *pqm-1* transcript and alter gene expression, particularly in the presence and absence of *C. elegans* ADARs, would be an important future direction to dissect the mechanism of *pqm-1* regulation in the nervous system.

Our results also demonstrate that while there is decreased expression of PQM-1 activated genes in L1 arrested *adr-2(-)* animals compared to wild type, there is no difference in gene expression when the L1-arrested animals are fed (Fig 1C). These data suggest that either another transcription factor drives the expression of the *dod* genes in fed, *adr-2(-)* L1 animals or that neural *pqm-1* expression does not decrease in fed L1 animals in the absence of *adr-2*. Our current data do not discriminate between these 2 possibilities. However, since our data also demonstrates that neural ADR-1 binding to *pqm-1* causes decreased *pqm-1* expression in *adr-2(-)* neural cells, we tested the possibility that ADR-1 binding to *pqm-1* was altered upon feeding. For this assay, the neural ADR-1 RIP assay was performed directly on hatched L1-arrested animals or after feeding for 6 h. On assessing *pqm-1* mRNA in the RIP assay, compared to the negative control, while there was almost a 3-fold enrichment of *pqm-1* mRNA in the neural ADR-1 RIP in the absence of *adr-2* in starved L1 animals, no enrichment for *pqm-1* mRNA in IPs was observed in fed animals of the same genetic background (S4 Fig). As neural ADR-1 binding to *pqm-1* impacts neural *pqm-1* expression in *adr-2(-)* L1-arrested animals and there is loss of ADR-1 binding upon feeding, these data support a model in which *pqm-1* expression in neural cells is altered upon feeding leading to changes in downstream gene expression. Interestingly, these data also suggest that ADAR binding to target RNAs can be influenced by environmental factors, which is an unexplored area and an exciting future direction.

As a previous study indicated heterodimer formation between ADR-1 and ADR-2 [46], studies have primarily focused on understanding how ADR-1 facilitates ADR-2 binding to mRNAs to promote editing [27,33,46]. However, several other studies have also revealed that ADR-1 competes with ADR-2 for binding certain mRNAs, which can lead to decreased editing

levels in transcripts [37,54]. Together, these studies suggest that the relationship between ADR-1 and ADR-2 is complex and may vary based on the tissue and developmental stage of the animals as well as on the individual transcript. In previous transcriptome-wide studies of ADR-1 mRNA binding, it was noted that ADR-1 binds nearly 1,200 transcripts in wild-type worms and while ADR-1 is bound to nearly 80% of these in animals lacking *adr-2*, ADR-1 also uniquely bound nearly 400 mRNAs in the absence of *adr-2* [35]. The impact of ADR-1 binding to these unique targets has not been investigated. However, as our study identifies important biological consequences of ADR-1 binding to *pqm-1* specifically in the absence of *adr-2*, the impact of ADR-1 binding should be explored further.

Our studies also revealed that loss of *adr-2* did not globally impact insulin signaling in neural cells, but instead, lead to further repression of genes negatively regulated in response to reduced insulin signaling (S2 Fig). Previous studies have indicated that the promoters of these genes contain an overrepresented sequence (CTTATCA), referred to as the DAE (DAF-16 associated element) [42] due to changes in expression of these genes upon loss of *daf-16* but lack of direct binding by DAF-16 [20]. The down-regulation of over 150 DAE-containing genes in neural cells lacking *adr-2* suggested a global regulator of these genes was altered. Previous studies have identified the transcription factor PQM-1 as an important factor regulating DAE-containing genes in L4 animals [20]. Herein, we found that loss of *adr-2* resulted in a neural-specific decrease in *pqm-1* expression, but decreased expression of the DAE-containing genes, *dod-17*, *dod-19*, and *dod-24* in both neural cells and the intestine. Transcription from the *pqm-1* promoter was observed in the nervous system and the intestine in early studies surveying transcription factor expression in *C. elegans* [55]. More recent studies reported that PQM-1:GFP translational fusions exhibited strong intestinal expression, but neural expression was not observed [20,56]. With these observations, it is not surprising that several recent studies have reported that PQM-1 has important impacts on intestinal gene expression [47,57]. However, consequences of loss of *pqm-1* specifically in neural cells have also been previously reported for transgenic animals that have altered proteostasis networks, specifically transcellular chaperone signaling [56]. Here, we have added to that body of work by identifying molecular changes that occur in neural cells with reduced *pqm-1* expression.

Our studies also indicate that *pqm-1* expression solely within the nervous system is sufficient to promote expression of *dod-17*, *dod-19*, and *dod-24* throughout the L1-arrested animal, providing the first evidence that PQM-1 can regulate gene expression in a cell non-autonomous manner. At present, it is unclear whether the cell non-autonomous regulation is dependent upon insulin signaling or if the downstream genes are affected by loss of *daf-2* [42] due to the impacts of DAF-2 on PQM-1 function in the nervous system. Our neural RNA-seq data did identify several ILPs, including *ins-4*, *ins-5*, *ins-26*, *ins-35*, and *daf-28* with significantly reduced expression in *adr-2(-)* neural cells (S5 Fig). Previous studies have shown that all of these ILPs are DAF-2 agonists [58–63]. Thus, it is possible that PQM-1 promotes expression of signaling molecules in the nervous system that relay information to the intestine to promote expression of the DAE-containing genes.

Another major unanswered question is what transcription factor could be mediating transcriptional control of the DAE-containing genes in the intestine. Previous studies of *C. elegans* transcription factors revealed an enrichment in DAE-containing sequences within the bound regions of 13 different proteins [20]. While PQM-1 was at the top of this list, it is possible that one of the other 12 transcription factors could be promoting transcription of DAE-containing genes in the intestine. Focusing on the promoters of *dod-17*, *dod-19*, and *dod-24* specifically, chromatin immunoprecipitation (ChIP) sequencing studies have identified binding sites for 3 transcription factors, PQM-1, FOS-1, and NHR-28 [64], but only FOS-1 and PQM-1 are present at the promoters of all 3 genes. FOS-1 expression and function in somatic gonad cells and

anchor cells is well established [65] and to date, there is no evidence that FOS-1 is expressed in intestinal cells. However, it is possible that in L1-arrested animals, FOS-1 expression changes and intestinal transcriptional activity could occur. In this same context, it is possible that other transcription factors occupy the *dod-17*, *dod-19*, and *dod-24* promoters within intestinal tissue to promote transcription during L1-arrest. Future studies should screen for factors needed specifically in the intestine for proper expression of DAE-containing genes; however, this may prove challenging in L1-arrested animals, where standard RNA interference (RNAi) by feeding cannot be employed.

Our work establishes a novel role for PQM-1 in survival of L1-arrested animals to $CoCl_2$ exposure. While a majority of L1-arrested wild-type animals die from acute exposure to high ($\geq$10 mM) doses of $CoCl_2$, *pqm-1(-)* animals exhibited significantly increased survival. It was also previously reported that loss of *pqm-1* resulted in increased survival of L4 animals exposed to low (5 mM) $CoCl_2$ for 20 h [47]. $CoCl_2$ is commonly used as a hypoxia mimetic, as it renders the prolyl hydroxylase, EGL-9, inactive, and allows for stabilization and activity of the transcription factor HIF-1 [66]. However, it has also been demonstrated that in *C. elegans*, $CoCl_2$ treatment can result in transcriptional responses that are both HIF-dependent and non-HIF mediated [48]. Furthermore, as $CoCl_2$ is also a heavy metal, it can induce toxicity through oxidative stress and mitochondrial fragmentation that is mainly regulated by the SKN-1 pathway [49]. Therefore, it is also important to verify effects observed with $CoCl_2$-mediated hypoxia to true hypoxia. The previous L4 study of PQM-1 function demonstrated that loss of *pqm-1* increased survival of L4 animals that were in a hypoxia chamber for 16 h followed by 24 h of normoxia. It would be important to perform similar studies for the L1 *pqm-1(-)* animals; however, the impacts of prolonged L1-arrest on *pqm-1* levels and ADAR regulation of those levels are unknown. At present, our data and the work of others suggests that PQM-1 is a negative regulator of hypoxic survival across developmental timescales, but it is unclear whether the cellular role of PQM-1 is the same in larval and adult animals.

Our phenotypic data revealed that PQM-1 function in neural cells is critical for its function as a negative regulator of hypoxic survival in L1-arrested animals. The previous study reported that PQM-1 promoted intestinal lipid levels and yolk protein transport to developing oocytes under oxygen depletion in adult animals [47]. However, it is important to note that direct binding of PQM-1 to the promoters of genes underlying the metabolic changes during adult hypoxic exposure was not demonstrated, thus it would be interesting to determine whether PQM-1 functions within the intestine or cell non-autonomously regulates hypoxic survival of adult animals.

At present, the physiological role for the negative regulation of survival by PQM-1 is unknown. Clearly on a cellular level, the ability to undergo metabolic changes that allow survival to hypoxia is a major aspect of oncogenesis; thus, players that keep this function in check are important. However, it is also well established in model organisms, including *C. elegans*, that adult animals reared under hypoxic conditions live longer than animals reared in normoxic conditions [67]. While promoting overall survival, the response to hypoxia is an energy-intensive process that disrupts cellular proteostasis [68]; thus, to preserve energy and maintain equilibrium, it is likely equally important to control levels of hypoxic responses. Interestingly, a recent study revealed that fasted animals have an altered response to hypoxia and that the DAF-2 pathway, independent of DAF-16, plays an important role in this response [69]. It would be interesting to see if the role we identified for PQM-1 in L1 animals hatched in the absence of food is also involved in the coordinated response of adult animals to nutritional state and hypoxia. Furthermore, as exposure to limiting oxygen or nutrients has been reported to have transgenerational effects on descendants' metabolic programming, behavior, and

fecundity in *C. elegans* [70–72], exploring the role of PQM-1 in these processes may shed light on the physiological role of negative regulation of hypoxic survival.

## Materials and methods

### *C. elegans* strains and maintenance

All worms were maintained under standard laboratory conditions on nematode growth media seeded with *Escherichia coli* OP50 [73]. The following previously generated strains were used in this study: Bristol strain N2, BB19 (*adr-1(tm668)*) [74], BB20 (*adr-2(ok735)*) [74], BB21 (*adr-1(tm668);adr-2(ok735)*) [74], HAH22 (*adr-2(gk777511)* [75] agIs6[*dod-24*::GFP] [43], *daf-2(m596)* [41], *pqm-1(ok485)* [45]. Neural cells were isolated from HAH45 (*prab3*::rfp:: *C35E7.6* 3′ UTR; *prab3*::gfp::*unc-54* 3′ UTR; *unc-119* genomic rescue), HAH46 (*adr-2(ok735)*; *prab3*::rfp::*C35E7.6* 3′ UTR; *prab3*::gfp::*unc-54* 3′ UTR; *unc-119* genomic rescue) and BB79 (*adr-1(tm668);adr-2(ok735)*; *prab3*::rfp::*C35E7.6* 3′ UTR; *prab3*::gfp::*unc-54* 3′ UTR; *unc-119* genomic rescue) [74].

Strains generated in this study include HAH23 (BB20 + blmEx18(*Y75B8A.8* 3′ UTR hairpin construct in *prab3*::GFP::*unc-54* 3′ UTR (pHH340); *prab3*::3XFLAG ADR-2 cDNA::*unc-54* 3' UTR (pHH438)), HAH24 (BB20 + blmEx19(*Y75B8A.8 3' UTR* hairpin construct in *prab3*:: GFP::*unc-54* 3' UTR Clone vector (pHH340))), HAH25 (BB19 + blmEx20(*prab3*::GFP::*unc-54* 3′ UTR (pHH21); *prab3*::3XFLAG ADR-1::*unc-54* 3′ UTR (pHH512))), HAH26 (BB21 + blmEx20(*prab3*::GFP::*unc-54* 3′ UTR(pHH21); *prab3*::3XFLAG ADR-1::*unc-54* 3′ UTR (pHH512))), HAH27 (*adr-2(ok735)*, agIs6[*dod-24*::GFP]), HAH28 (BB20 + blmEx18 (*Y75B8A.8* 3′ UTR hairpin construct in *prab3*::GFP::*unc-54* 3′ UTR (pHH340); *prab3*::3X-FLAG ADR-2 cDNA::*unc-54* 3′ UTR (pHH438), agIs6[*dod-24*::GFP]), HAH29 (*adr-2(ok735)*, *daf-2(m596)*), HAH30 (wild type), HAH31 (*daf-2(m596)*), HAH32 (*adr-2(ok735)*), HAH33 (*adr-2(ok735)*, *daf-2(m596)*), HAH42 (HAH38+ blmEx19(*Y75B8A.8* 3′ UTR hairpin construct in *prab3*::GFP::*unc-54* 3′ UTR (pHH340))), HAH43 (HAH38+blmEx21(*Y75B8A.8* 3′ UTR hairpin construct in *prab3*::GFP::*unc-54* 3′ UTR (pHH340); *prab3*::pqm-1::*unc-54* 3′ UTR (pHH549))), HAH47 (3X FLAG ADR-1 dsRBD1 (K223E, K224A, K227A) CRISPR).

Animals created by microinjection (HAH23-HAH26, HAH28, HAH40-HAH44) used standard microinjection techniques and were passaged by selecting worms that contained the GFP co-injection marker. The injection mix contained 20 ng/µl of the co-injection marker and 1 ng/µl of the transgene of interest.

Animals created by CRISPR modification (HAH47) used standard microinjection techniques and were identified as rolling F1 progeny and non-rolling F2 progeny. Injection mix for the ADR-1 dsRBD1 mutant strain included 1.5 µM Cas9 (IDT, Alt-R Cas9 nuclease V3), 4 µM tracrRNA (IDT), 4 µM of crRNA (IDT) (HH3088) (S2 Table), 37 ng/µl *rol-6* plasmid (HAH293), and 4 µM of repair template ssODN (HH3089) (S2 Table) containing the mutation in ADR-1 dsRBD1(KKxxK-EAxxA). Genomic modifications were verified using Sanger sequencing and ADR-1 expression was verified using western blot.

Crosses were performed by putting 9 to 10 males and 1 hermaphrodite on mating plates and genotyping was performed for the F1 progeny and F2 progeny using primers mentioned in S2 Table. The specific crosses performed included: creation of HAH27 by crossing agIs6 [*dod-24*::GFP] hermaphrodites to BB20 males, creation of HAH28 by crossing HAH27 hermaphrodites to HAH23 males, creation of HAH29 by crossing *daf-2(m596)* hermaphrodites to BB20 males, creation of HAH34 by crossing *pqm-1(ok485)* hermaphrodites to N2 males, creation of HAH35 (wild type), HAH37 (*adr-2(ok735)*), HAH38 (*pqm-1(ok485)*), HAH39 (*adr-2 (ok735); pqm-1(ok485)*) by crossing HAH34 hermaphrodites to BB20 males, creation of HAH40 (BB20 + blmEx19(*Y75B8A.8* 3′ UTR hairpin construct in *prab3*::GFP::*unc-54* 3′ UTR

(pHH340))) and HAH41 (blmEx19(*Y75B8A.8* 3′ UTR hairpin construct in *prab3*::GFP::*unc-54* 3′ UTR (pHH340))) by crossing HAH24 hermaphrodites to N2 males, creation of HAH44 (BB20+ blmEx21(*Y75B8A.8* 3′ UTR hairpin construct in *prab3*::GFP::*unc-54* 3′ UTR (pHH340); *prab3*::*pqm-1*::*unc-54* 3′ UTR (pHH549))) by crossing HAH43 hermaphrodites to BB20 males, creation of HAH48 (wild type), HAH49 (*adr-2(ok735)*), HAH50 (*adr-2(ok735)*; ADR-1 dsRBD1 (KKxxK-EAxxA) CRISPR) by crossing HAH47 hermaphrodites to BB20 males and HAH57 (*adr-2(ok735)*; ADR-1 dsRBD1 (KKxxK-EAxxA) CRISPR; *prab3*::*rfp*:: *C35E7.6* 3′ UTR; *prab3*::*gfp*::*unc-54* 3′ UTR; *unc-119* genomic rescue) by crossing BB79 hermaphrodites to HAH50 males.

## Cloning

To generate the neural ADR-2 complementary DNA (cDNA) expressing animals, the *adr-2* cDNA sequence was amplified from a plasmid with primers HH1962 and HH1963 (Table S2). This fragment was cut with restriction enzymes BglII and SalI and then cloned into plasmid pHH326 (*prab3*::GFP::*unc-54* 3′ UTR) to generate a plasmid expressing neural *adr-2* cDNA. The sequence of the *adr-2* region cloned into the plasmid was confirmed using Sanger sequencing.

To generate the neural ADR-1 expressing animals, the *rab3* promoter sequence was amplified from plasmid pHH326 (*prab3*::GFP::*unc-54* 3′ UTR) with primers HH170 and HH2771 (S2 Table). This fragment was cloned into plasmid pHH99 (pBluescript 3X FLAG genomic *adr-1*) previously published [33], with restriction enzymes KpnI and PstI to generate plasmid pHH512 with *prab3*::3XFLAG ADR-1. The sequence of the *rab3* promoter cloned into the plasmid was confirmed using Sanger sequencing.

The vector used to express *pqm-1* in the nervous system, pWorm[Exp]-rab-3>cel_pqm-1, was constructed by VectorBuilder. The vector ID is VB221230-1025zug, which can be used to retrieve detailed vector information from vectorbuilder.com.

## Gene set enrichment analysis

Gene set enrichment analysis was performed using the online WormCat software [38] for up-regulated and down-regulated genes identified from comparing the wild-type and *adr-2(-)* neural RNA-seq datasets.

## Bleaching

Synchronized first larval stage (L1) animals were obtained by bleaching with 5 M NaOH and Clorox solution. After bleach solution was added, animals were incubated on a shaker at 20°C for 7 min and then spun down to collect embryos. Collected embryos were washed with 1× M9 buffer (22.0 mM KH$_2$PO$_4$, 42.3 mM Na$_2$HPO$_4$, 85.6 mM NaCl, 1 mM MgSO$_4$) solution thrice. The animals were incubated overnight in 1× M9 solution at 20°C. Next day, hatched L1 worms were spun down and washed again with 1× M9 solution thrice.

## Neural cell isolation and COPAS sorting

Neural cells were isolated from synchronized first larval stage worms as previously described [27] and filtered into sterile FACS tubes. Briefly, staining with near IR live/dead fixable dye (Invitrogen) of the isolated neural cells was done before performing FACS sorting. The BD FACSAria II sorter was used to separate the GFP+ neural cells from the non-GFP cells, and FACSDiva 6.1.1 software was used to analyze the sort (IU Flow Cytometry Core Facility). Sorted neural cells were collected into conical tubes with TRIzol (Invitrogen), snap-frozen in

liquid nitrogen, and stored at −80°C. For sorting transgenic animals, the COPAS BioSelect instrument (IU Flow Cytometry Core Facilty) was used to isolate GFP+ animals based on Time of Flight (TOF) and Extinction (Ext), and 250 transgenic GFP+ animals were sorted per strain and collected on unseeded 10 cm NGM plates.

## Bioinformatics analysis for differential gene expression

N2 and *adr-2(-)* L1 neural datasets generated in [37] under accession number GSE1151916 were downloaded and analyzed, and 75 bp single-end stranded RNA-sequencing reads were subjected to adaptor trimming and aligned to the *C. elegans* genome (WS275) using STAR (v2.7.8a) with the parameters: [*runThreadN 8*, *outFilterMultimapNmax 1*, *outFilterScoreMinOverLread 0.66*, *outFilterMismatchNmax 10*, *outFilterMismatchNoverLmax: 0.3*]. Indexing of the aligned bam files was performed using samtools (v1.3.1) and featureCounts (v2.0.1) was used to generate the raw read counts file. DESeq2 library (v1.26.0) on R studio [76] was used to process the raw read counts and generate the counts.csv file used for differential gene expression analysis.

## RNA isolation and quantitative real-time PCR (qPCR)

RNA extraction was performed using TRIzol (Invitrogen) reagent and DNA contamination was removed by treatment with TURBO DNase (Ambion) followed by the RNeasy Extraction kit (Qiagen) and stored at −80°C. Concentrations of the RNA samples and presence of any contamination with organic and protein components was determined using a Nanodrop (Fisher Scientific). For qPCR experiments using RNA from L1 animals, 2 μg of DNase-treated RNA was reverse transcribed into cDNA using Superscript III (Invitrogen) with random hexamers (Fisher Scientific) and oligo dT (Fisher Scientific) primers. For qPCR experiments using RNA from neural cells or COPAS sorted transgenic animals, the whole 12 μl was reverse transcribed into cDNA. Following reverse transcription of RNA from L1 animals, 20 μL of water was added to the cDNA. For RIP experiments, 200 ng RNA for inputs and the whole 12 μl of IP samples was reverse transcribed into cDNA and no water was added to the cDNA. Gene expression was determined using SybrFast Master Mix water and gene-specific primers (S2 Table) on a Thermo Fisher Quantstudio 3 instrument. The primers designed for qPCR (S2 Table) spanned an exon-exon junction to prevent detection of genomic DNA in the samples. Melting curves were generated for all primer pairs used to ensure high quality of qPCR products. For each gene analyzed, a standard curve of 8 to 10 samples of 10-fold serial dilutions of the amplified product were used to generate a standard curve of cycle threshold versus the relative concentration of amplified product. Standard curves were plotted on a logarithmic scale in relation to concentration and fit with a linear line. Fit ($r^2$) values were around 0.99 and at least 7 data points fell within the standard curve. Each cDNA measurement was performed in 3 technical replicates, and each experiment was performed in 3 biological replicates.

## Fluorescence microscopy

Synchronized L1 animals were anesthetized on agarose pads containing sodium azide followed by a coverslip. Images were taken using the Leica SP8 Scanning Confocal Microscope (IU Light Microscopy Imaging Core) and the 10× objective. Each image was taken with the intestinal cells and head ganglia region in focus. For each trial, exposure time was calibrated to minimize the number of saturated pixels for that set of animals. The Fiji software was used to quantify the fluorescence intensities in the animals as measured by intensity of each pixel in the selected area of a frame (i.e., the worm). The fluorescence intensity for the desired region was first determined by outlining the region and obtaining the area and integrated density

values. For increased accuracy, 3 measurements were taken for each area and the mean fluorescent intensity was obtained. The corrected fluorescence intensity was calculated by subtracting out the mean background signal from the mean fluorescent intensity for all the animals imaged.

## RNA immunoprecipitation

Synchronized L1 worms were washed with IP buffer (50 mM HEPES [pH 7.4], 70 mM K-Acetate, 5 mM Mg-Acetate, 0.05% NP-40, and 10% glycerol) containing a mini EDTA-free cOmplete protease inhibitor tablet (Roche) and UV crosslinked (3 J/cm2) using the Spectrolinker (Spectronics). The worms were then frozen into pellets using liquid nitrogen and stored at −80°C. The frozen worm pellets were ground on dry ice with a cold mortar and pestle, and the cell lysate was centrifuged at maximum speed for 10 min to remove cellular debris. To collect L1 worms after feeding, synchronized L1 animals were plated on NGM plates seeded with normal bacterial food for 6 h and then, washed off the plate with 1× M9 buffer thrice before washing with IP buffer as described above. Protein concentration was measured using Bradford reagent (Sigma) and the entire L1 lysates were added to 25 μl magnetic Protein-G beads (Invitrogen) for preclearing. After incubation for 1 h on a rotator at 4°C, 500 μg of the L1 lysate from the supernatant was added to 25 μl anti-FLAG magnetic beads (Sigma). After incubation for 1 h on a rotator at 4°C, protein-bound beads were washed with wash buffer thrice (0.5 M NaCl, 160 mM Tris-HCl [pH 7.5], 0.1% NP-40, 0.25% Triton X-100) containing a mini EDTA-free cOmplete protease inhibitor tablet (Roche). A portion of the IP (2/5) was stored in 2× SDS loading buffer and used for immunoblotting. The remaining beads were incubated with 1 μl RNasin (Fisher) and 0.5 μl of 20 mg/ml Proteinase K (NEB) at 42°C and 1,200 rpm for 15 min in a thermomixer. RNA was isolated and reverse transcription was performed using random hexamer primers. Following reverse transcription, qPCR was performed as described above. An overview of this methodology, analysis and trouble-shooting can also be found in [77].

## Immunoblotting

Protein lysates were prepared as mentioned above. Protein lysates were boiled for 5 min and 100 μg of IP lysates and 10 μg of input lysates were subjected to SDS-PAGE. The same immunoblot was treated with antibodies against FLAG (Sigma, M8823) and β-Actin (Cell Signaling, 8457S) after cutting the blot. Protein bands were visualized using enhanced chemiluminescent detection SuperSignal West Femto Maximum Sensitivity Substrate (Fisher). The immunoblot images without saturation were acquired using Image Lab software (version 6.1.0 build 7) in the BIO-RAD ChemiDoc MP imaging system.

## Cobalt chloride exposure assays

These assays were performed as previously described [49] with slight modifications.

Briefly, cobalt(II) chloride hexahydrate (CoCl$_2$) powder (Sigma-Aldrich) was used to make a 0.1 M stock solution with distilled water and then working stocks of varying concentrations were prepared in 85 mM NaCl. Bleaching according to above mentioned conditions was performed to obtain L1-arrested animals. After washing the hatched animals twice with 85 mM NaCl, 5,000 L1 animals per strain were exposed to CoCl$_2$ of varying concentrations in a total volume of 500 μl for 2 h. After exposure, the worms were washed twice with 85 mM NaCl to remove any residual CoCl$_2$ and other debris. After washing, 40 worms per strain were transferred to 35 mm NGM plates seeded with OP50 and incubated at 20°C for 24 h. After the incubation, alive and dead worms were counted in triplicates for each strain and 3 biological

replicates for both the transgenic and non-transgenic sets of worms. Genotypes were blinded before the bleaching to reduce bias.

## Statistical analysis

The data for all experiments was plotted and analyzed using the recommended statistical test on GraphPad Prism. The statistical test used is mentioned in the figure legends.

## Supporting information

**S1 Fig. Gene set enrichment analysis for genes with altered expression in *adr-2(-)* neural cells compared to wild-type neural cells.** For the total number of input genes in each of the categories (regulated gene set), the *P* value is calculated using Fisher's exact test. "Count" indicates the number of genes within a specific category. The size and color of the circles for each of the categories signifies the number of genes (size) and *P* value (color) for the categories mentioned (see key in figure).
(TIFF)

**S2 Fig. Overlap between datasets from Tepper and colleagues and genes misregulated in *adr-2(-)* neural cells.** DAF-16 activated (yellow circle) and PQM-1 activated (blue circle) genes from Tepper and colleagues were individually overlapped with the genes found to be up-regulated (green circle) and down-regulated (light red circle) genes in *adr-2(-)* neural cells compared to wild-type neural cells. The number in parentheses denotes the number of over-lapped genes between the 2 datasets expected due to random chance.
(TIFF)

**S3 Fig. L1 survival under hypoxic conditions.** (A, B) Survival of transgenic (A) and non-transgenic (B) hatched L1 animals with number of alive L1s on Y axis and varying concentrations of $CoCl_2$ on X axis. Error bars represent standard error of the mean (SEM). Statistical significance across strains was calculated using two-way ANOVA test. ****$p < 0.0001$. All individual data and statistics are included in S5 Data under Supporting information.
(TIFF)

**S4 Fig. Neural ADR-1 binds *pqm-1* in the absence of *adr-2* specifically in starved L1 animals.** Bar graph represents the fold enrichment determined by dividing the IP/Input value from qPCR for the indicated strains divided by that of negative control. The mean of 3 biological replicates was plotted. Error bars represent SEM. Statistical significance was calculated by multiple unpaired *t* tests followed by Holm–Šídák multiple comparisons correction. *****$p < 0.000001$. All individual data and statistics are included in S6 Data under Supporting information.
(TIFF)

**S5 Fig. DAF-2 ligands down-regulated in *adr-2(-)* neural cells.** Plot depicting expression of the 20 moderately expressed (read counts between 50 and 300) or highly expressed (read counts >300) ILPs in *adr-2(-)* neural cells from the neural RNA sequencing dataset. Red dots indicate ligands that have significantly decreased expression in *adr-2(-)* neural cells compared to wild-type neural cells and are annotated. ILPs with *p* value <0.05 and log2fold change <-0.5 were considered significantly down-regulated. All individual data and statistics are included in S7 Data under Supporting information.
(TIFF)

**S1 Table. Differentially expressed genes in *adr-2(-)* neural cells compared to wild-type neural cells (related to Fig 1).** Genes differentially expressed in L1 *adr-2(-)* neural cells compared

to L1 wild-type neural cells with a *p* value <0.05 and log2fold change >|0.5|. Expression of all genes in the neural RNA sequencing datasets from 3 biological replicates is listed by Wormbase Annotation (column A) and gene name (Column B). log2fold change values, raw *p* values and the adjusted *p* values as obtained from the DESeq2 analysis are shown in columns C, D, and E, respectively. Read counts from the DESeq2 analysis for the 3 biological replicates of *adr-2(-)* neural cells are listed in columns F,G, and H. Read counts from the DESeq2 analysis for the 3 biological replicates of wild-type neural cells are listed in columns I,J, and K. (XLS)

**S2 Table. Sequences of all primers used in this study for qPCR, reverse transcription, genotyping, PCR amplification, and CRISPR.**
(XLSX)

**S1 Data. Individual quantitative observations for results illustrated in Fig 1.**
(XLSX)

**S2 Data. Individual quantitative observations for results illustrated in Fig 2.**
(XLSX)

**S3 Data. Individual quantitative observations for results illustrated in Fig 3.**
(XLSX)

**S4 Data. Individual quantitative observations for results illustrated in Fig 4.**
(XLSX)

**S5 Data. Individual quantitative observations for results illustrated in Figs 5 and S3.**
(XLSX)

**S6 Data. Individual quantitative observations for results illustrated in S4 Fig.**
(XLSX)

**S7 Data. Individual quantitative observations for results illustrated in S5 Fig.**
(XLSX)

**S1 Raw Images. Raw western blot images for all biological replicates for results illustrated in Fig 4B.**
(PDF)

**S2 Raw Images. Raw western blot images for all biological replicates for results illustrated in S4 Fig.**
(PDF)

## Acknowledgments

We thank Christiane Hassel (IUB- Flow Cytometry Core Facility) for assisting in COPAS sorting of transgenic animals and isolation of neural cells. We thank Dr. Andras Kun (IUB- Light Microscopy Imaging Center) for the training and facilitating usage of the confocal microscope. We thank current members of the Hundley lab, Dr. Chinnu Salim, Boyoon Yang, Emily Erdmann, Mary Skelly, and former Hundley lab member Dr. Reshma Raghava Kurup for careful reading of the manuscript. We thank graduate student Shefali Shefali for her tremendous help in taking the confocal images and Boyoon Yang for assisting in the masking of genotypes for the survival assays.

## Author Contributions

**Conceptualization:** Ananya Mahapatra, Alfa Dhakal, Heather A. Hundley.

**Data curation:** Ananya Mahapatra, Aika Noguchi.

**Formal analysis:** Ananya Mahapatra, Aika Noguchi.

**Funding acquisition:** Ananya Mahapatra, Heather A. Hundley.

**Investigation:** Ananya Mahapatra.

**Methodology:** Ananya Mahapatra, Aika Noguchi.

**Project administration:** Ananya Mahapatra, Heather A. Hundley.

**Resources:** Alfa Dhakal, Pranathi Vadlamani, Heather A. Hundley.

**Supervision:** Heather A. Hundley.

**Validation:** Ananya Mahapatra.

**Writing – original draft:** Ananya Mahapatra, Heather A. Hundley.

**Writing – review & editing:** Ananya Mahapatra, Alfa Dhakal, Heather A. Hundley.

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
