## [Editor Report · Decision Letter 0]

3 May 2023

Dear Dr Hundley, 

Thank you for submitting your manuscript entitled "ADARs employ a neural-specific mechanism to regulate PQM-1 expression and survival from hypoxia" for consideration as a Research Article by PLOS Biology.

Your manuscript has now been evaluated by the PLOS Biology editorial staff, as well as by an academic editor with relevant expertise, and I am writing to let you know that we would like to send your submission out for external peer review.

Once your full submission is complete, your paper will undergo a series of checks in preparation for peer review. After your manuscript has passed the checks it will be sent out for review. To provide the metadata for your submission, please Login to Editorial Manager (https://www.editorialmanager.com/pbiology) within two working days, i.e. by May 05 2023 11:59PM.

Kind regards,

Richard

Richard Hodge, PhD

Associate Editor, PLOS Biology

rhodge@plos.org

PLOS

---

## [Decision Letter · Decision Letter 1]

13 Jun 2023

Dear Dr Hundley,

Thank you for your patience while your manuscript "ADARs employ a neural-specific mechanism to regulate PQM-1 expression and survival from hypoxia" was peer-reviewed at PLOS Biology. Please accept my apologies for the delays that you have experienced during the peer review process. Your manuscript has now been evaluated by the PLOS Biology editors, an Academic Editor with relevant expertise, and by three independent reviewers. 

In light of the reviews, which you will find at the end of this email, we would like to invite you to revise the work to thoroughly address the reviewers' reports.

As you will see, Reviewers #1 and #2 are generally positive and think the manuscript is interesting and well-done. They both note that the expression of PQM-1 RNA upon ADR1/2 deletion should be assessed and that summary statistics for the microscopy images are provided. Reviewer #3 raises concerns with the normalization of several pieces of data and that the manuscript does not provide sufficient evidence to show that the effects are specific to the daf-2 pathway. 

Given the extent of revision needed, we cannot make a decision about publication until we have seen the revised manuscript and your response to the reviewers' comments. Your revised manuscript is likely to be sent for further evaluation by all or a subset of the reviewers.

**IMPORTANT - SUBMITTING YOUR REVISION**

*Re-submission Checklist*

*Published Peer Review*

*PLOS Data Policy*

*Blot and Gel Data Policy*

Sincerely,

Richard

Richard Hodge, PhD

rhodge@plos.org

REVIEWS:

Reviewer #1: In this manuscript, Mahapatra et al., evaluate the role of C. elegans ADAR proteins (ADAR-1 and ADR-2) in regulating the insulin signaling pathway. This work was based on previous findings from the Hundley Lab in which they observed A-to-I editing of the daf-2 mRNA. Since DAF-2 is the only insulin receptor in C. elegans, Mahapatra et al., sought to determine if A-to-I editing is important in regulating DAF-2 and insulin signaling, with a focus on the nervous system in L1-arresested animals. The authors use a series of genetic experiments to show that loss of adr-2 (the A-to-I editing enzyme of C. elegans) caused reduced expression of genes downstream of insulin signaling. Loss of adr-2 phenocopied loss of daf-2. Interestingly, this phenotype could be rescued by feeding the animals that had been arrested in L1 by hatching in the absence of food. Expression of neuronal ADR-2 was also capable of rescuing the phenotype. Additionally, expression of an editing deficient ADR-2 was also capable of rescuing the phenotype, indicating that ADR-2 regulates insulin signaling without A-to-I editing. Guided by their RNA-seq data, the authors focused on the transcription factor PQM-1 which is downstream of DAF-2. Loss of pqm-1 phenocopied loss of adr-2, and loss of adr-2 caused reduced expression of pqm-1. Given that ADR-2 dimerizes with ADR-1 and can influence which RNAs ADR-1 interacts with, the authors next turned to ADR-1 to access its role in this pathway. Knockout of adr-1 could rescue the phenotype of adr-2 loss. Furthermore, the dsRNA binding ability of adr-1 was required for the reduced insulin signaling observed upon adr-2 loss. The authors show that adr-1 binds to the pqm-1 RNA, but only in the absence of adr-2. Based on this finding, the authors generate a model that adr-1 binding to pqm-1 RNA following loss of adr-2 causes reduced expression of PQM-1. Finally, given PQM-1s role in regulating survival following hypoxic stress, the authors next explored the importance of adr-1 and adr-2 in survival from hypoxia. The authors used CoCl2 treatment as a proxy for hypoxia and showed reduced survival following treatment for animals lacking adr-2 or pqm-1. This phenotype could be rescued by neural expression of PQM-1 or ADR-2, and by mutation of the dsRBD of ADR-1. Overall, these findings show an important role for ADR-1 and ADR-2 in regulating insulin signaling and survival from hypoxia in C. elegans, and highlight an interesting regulatory mechanism whereby presumably binds and destabilizes an RNA in the absence of ADR-2. 

Major concerns:

1. The authors didn't access pqm-1 RNA expression in the experiments that generated Figure 4A and 4C. Specifically, does pqm-1 RNA expression remain the same as WT when both adr-1 and adr-2 are deleted (as does expression of dod-17, dod-19 and dod-24). Or for panel C, does the loss of ADR-1 dsRNA binding ability prevent reduced expression of pqm-1 RNA in the absence of adr-2. These data would greatly support the rest of the manuscript.

2. The authors show that feeding the L1 arrested animals can rescue the phenotype of adr-2 loss. This raises a question of how that influences regulation of pqm-1 RNA by ADR-1. In the discussion the authors suggest that feeding could influence ADR-1 binding to pqm-1 RNA (which could be tested). One possibility is that after feeding another transcription factor (DAF-16) is driving expression of DOD-17 DOD-19 and DOD-24. The authors could assess PQM-1 expression as they did for the other genes in Figure 1C. Is PQM-1 expression rescued by feeding? Or possibly ADR-1 expression is reduced upon feeding? There are several possibilities here that the authors should discuss further (in a separate paragraph) and some experiments that could be done to strengthen the manuscript. 

Minor concerns:

1. Is it possible that the CoCl2 treatment used as a hypoxia mimic be activating the insulin/daf-2/pqm-1 pathway via another means? Given Co is a heavy metal, it may be activating other stress pathways leading to the observed phenotypes. Is there another way of doing this experiment, or an additional control that could be added to address this concern?

2. The authors use a two-way ANOVA for several analysis where this reviewer doesn't think it is the appropriate test. Typically, that test would be used for experiments with two variables that are changing (like genotype (WT vs. Mut.) and concentration of a drug (0,1,10, etc.)). That doesn't seem to be the case in places where it is used. Furthermore, it would most likely need a post-hoc test to identify which comparisons are significant.

3. The authors use 'multiple t-tests' for some comparisons but don't describe which method of multiple comparison correction they are using. 

4. The description for Figure 2B is lacking. It would also be nice to include summary statistics for all of the animals imaged. 

Reviewer #2: In C. elegans, ADR-2 is the only active ADAR enzyme. One of the targets of ADAR2 is daf-2. Interestingly daf-2 and adr-2 mutants display similar phenotypes, raising the question whether the editing of daf-2 is directliy linking to insulin signaling.

In a series of elegant experiments the authors show convincingly, that targets ot daf-2 are indeed downregulated upon loss of adr-2, while daf-2 itself seems not affected. Next, the authors test whether there is a link betwen the nervous system and the intestine. Indeed, the authors show that targeted expression of adr-2 in the nervous system can restore dod-24 promoter activity in the intestine. By expressing editing deficient mutant versions of ADR-2, the authors show that editing activity is not required to activate the insulin activation pathway.

In dissecting the regulator of DAF-2 regulated genes, the authors identify PAM-1 genes, a direct target of DAF-2 to be downregulated upon adr-2 deficiency. Interestingly, this finding is restricted to the nervous system. Consistently, restoring expression of pqm-1 in the nervous system is sufficient to overcome adr-2 deficiency. In trying to unravel the underlying mechanisms, the authors discover that ADR-1 is normally bound by ADR-2, thereby preventing its binding to pqm-1 RNA. In the absence of ADR-2, however, ADR-1 binds pqm-1 RNA, leading to downregulatio of downstream genes.

Overall, the authors present a well-structured and readable manuscript. The experiments are well performed and documented. Thus, this research provides insights into how the nervous system adapts to environmental cues and promotes survival in hypoxic conditions through post-transcriptional gene regulation

However, prior to publication two points should be addressed.

1) Figure 2B shows only one organism in the field. It would be important to provide this micrograph with some statistics on more worms.

2) Can the authors speculate what is happening to pqm-1 RNA in the absence of ADR-1? Does pqm 1 contain a structure element that might serve as a target for a dsRBD? Is translation affected, leading to a destablization of pqm-1? Are there experiments testing for ribosome profiling or RNA stability in the presence or absence of adr-1.?

Reviewer #3: Mahapatra et al. present a manuscript exploring links between RNA binding proteins, stress responses and aging. One on the RNA binding proteins in the study, was previously shown to be important for RNA editing of daf-2. This study set out to test the hypothesis that adr-2, acting in neuronal cells, could impact daf-2 related insulin signaling pathways. They use gene regulation from RNA seq, genetic assays and studies of expression of daf-16 regulated genes to conclude that expression of a transcription factor, pqm-2, is regulated by dual binding of these RNA binding proteins (ADR-1 and ADR-2) and that the neuronal expression is key for stress responses. However, there are many pathways that affect aging and this study doesn't not address any alternative hypotheses to show that effects are specific to the daf-2 pathway. The test genes used, dod-17, -19 and -24 do not appear to have any biological relevance to the pathway and while they are regulated by daf-2, wormbase each of these genes change in more than 200 biological contexts, many of them unrelated to insulin signaling. Thus, they can not be used as a proxy for insulin pathway stimulated actions and enthusiasm for the study in the present form is limited.

1. The authors show RNA seq data from purified neurons. While they cite this RNA seq from a separate study, it would be important to show controls for the specificity of the purification. More importantly, how do these gene compare with daf-2 regulated genes, or genes regulated by other lifespan-regulating pathways such as the TGF-B pathway, mitochondrial mutants or the folate cycle? This type of analysis would be essential for linking ard-2 and insulin signaling. It is not sufficient to say that sharing life-span extension shows a connection. WormCat shows only very general categories, are there more specific categories that are also enriched?

2. The rationale for using the three dod genes appears weak. These genes appear to be decreased in both pure neuronal and entire animal qPCR (Fig1 B and C), therefore these effects appear to be occurring generally. Are these genes the most highly regulated, or were they cherry picked because they are co-regulated? The conclusion for Fig 1D is that the "expression profile is similar", however, the data simply shows that three genes require either ard-2 or daf-2 for expression. 

3. The data in Figure 4A is insufficient to support one of the major conclusions. The authors use qPCR to compare levels of the three dod genes in ard-1 and -2 mutants with those lacking both genes. The authors conclude that ard-1 has unique functions, as the qPCR values were significant as opposed to adr-2. However, the ard-2 values had a very high standard deviation, thus, the lack of a significance is highly likely to be technical and not biological.

4. Figure 4B shows an RNA precipitation comparing binding of a flag tagged ard-2 transgene. There are several points which make interpretation of this data difficult. First, it is normalized to total RNA and completed in whole animal lysates, therefore neuronal specific effects would be difficult to separate. The normalization to the RNA values in the qPCR appears to provide the baseline in the quantitation (since no protein comes down). This seems difficult to call a baseline value for protein measurement solely on RNA levels, unless the description of the method is unclear. From the blot, it appears the IP'd amounts are quite similar and a lower input amount is what provides the increase in the quantitation. However, with the dynamics of RNA protein interactions, shouldn't the input values be the same, then the differences measured? 

5. The normalization of Figure 4C is done differently than that other assays. Samples are normalized to the mutant, rather than the control. While the authors conclude that a significant increase occurred in the ard-2(-); ARD-1 dsRBD mutant, however, there is no ard-1; ard-2 shown for direct comparison and if the data were normalized to the control, the graph would appear to show that the dsRBD1 mutant simply returns to wild type levels. 

6. The conclusions from Figure 5 appear inconsistent with the data. In both figures, loss of pqm-1 or adr-2 confer a survival advantage. Expression of neural pqm-1 is insufficient to provide this advantage. To make the conclusion that this effect occurs in the neurons, you would need to show neuronal only expression conferred a different response than loss of expression. In B, there is no adr-2; adr-1 without the RBD as a control. Furthermore, overexpressing transcription factors is prone to artifacts, as they can bind lower affinity sequences.

---

## [Decision Letter · Decision Letter 2]

9 Aug 2023

Dear Dr Hundley,

Thank you for your patience while we considered your revised manuscript "ADARs employ a neural-specific mechanism to regulate PQM-1 expression and survival from hypoxia" for consideration as a Research Article at PLOS Biology. Your revised study has now been evaluated by the PLOS Biology editors, the Academic Editor and the original reviewers.

The reviews are attached below. As you can see, Reviewers #1 and #2 (who have expertise in RNA editing) are now satisfied with the revision and the additional data included on PQM-1 expression. However, Reviewer #3 (who has expertise in C. elegans and metabolism) still raises concerns with the overall strength of the evidence to support the conclusions implicating insulin signalling in the model and requests that some additional analyses are included to demonstrate specificity. In addition, we ask that you please include Response Figure 2 in the main manuscript, as it reports a control for Figure 4D. 

In light of the reviews, we are pleased to offer you the opportunity to address the comments from Reviewer #3. We will then assess your revised manuscript and your response to the reviewers' comments with our Academic Editor aiming to avoid further rounds of peer-review, although might need to consult with the reviewers, depending on the nature of the revisions.

I would also be grateful if you could make sure to address the following data and other policy-related requests that I have provided below (A-F):

(A) We would like to suggest the following modification to the title: 

“ADAR-mediated regulation of PQM-1 expression in neurons impacts gene expression throughout C. elegans and regulates survival from hypoxia”

(B) You may be aware of the PLOS Data Policy, which requires that all data be made available without restriction: http://journals.plos.org/plosbiology/s/data-availability. For more information, please also see this editorial: http://dx.doi.org/10.1371/journal.pbio.1001797

-Supplementary files (e.g., excel). Please ensure that all data files are uploaded as 'Supporting Information' and are invariably referred to (in the manuscript, figure legends, and the Description field when uploading your files) using the following format verbatim: S1 Data, S2 Data, etc. Multiple panels of a single or even several figures can be included as multiple sheets in one excel file that is saved using exactly the following convention: S1_Data.xlsx (using an underscore).

-Deposition in a publicly available repository. Please also provide the accession code or a reviewer link so that we may view your data before publication. 

Figure 1A-D, 2A-C, 3A-D, 4A-D, 5A-B, S3A-B , S4, S5

(C) Please also ensure that each of the relevant figure legends in your manuscript include information on *WHERE THE UNDERLYING DATA CAN BE FOUND*, and ensure your supplemental data file/s has a legend.

(D) We require the original, uncropped and minimally adjusted images supporting all blot and gel results reported in the following Figures:

Figure 4B, S4

We note that Figure 4B has already been included in the Raw Images file, but this document is missing Figure S4. We will require these files before a manuscript can be accepted so please prepare and upload them now. Please carefully read our guidelines for how to prepare and upload this data: https://journals.plos.org/plosbiology/s/figures#loc-blot-and-gel-reporting-requirements

(E) Please ensure that your Data Statement in the submission system accurately describes where your data can be found and is in final format, as it will be published as written there. 

(F) Please note that per journal policy, the model system/species studied should be clearly stated in the abstract of your manuscript. 

**IMPORTANT - SUBMITTING YOUR REVISION**

*Resubmission Checklist*

*Published Peer Review*

*PLOS Data Policy*

*Blot and Gel Data Policy*

Sincerely,

Richard

Richard Hodge, PhD

rhodge@plos.org

REVIEWS:

Reviewer #1: I appreciate the authors efforts to address my concerns. The additional experiment showing binding of ADR-1 binding to pqm-1 is influenced by starving is very interesting. I look forward to seeing future work exploring how starvation influences ADR-1 binding. 

Reviewer #2: In this revised version of their manuscript, the authors address all comments raised by me. In particular, for figure 2, a statistics on the GFP fluorescence of pdod::GFP is presented.

Further, while the mechanistic consequences of ADR-1 binding to pqm-1 RNA are still not understood, the authors provide a lengthy discussion on the potential mechanisms and also provide related background information that was missing in the original version of the manuscript. 

Reviewer #3: Mahapatra et al present a revised manuscript exploring links between neuronal expression of RNA editing enzymes, expression of a stress responsive gene and treatment with cobalt chloride. In my initial review, I had concerns about overly broad interpretations of what the assays used in this study can conclude. In most cases, the authors have not provided sufficient rationale. The dod genes and pqm genes are being used as sensors. As their biological functions are not clear, it is important that the specificity of the regulation is rigorously shown, if not the premise of the study is flawed. 

1. The authors use several lines of evidence to implicate insulin signaling, however, insulin signaling was not shown to be either necessary or sufficient to the effects in the study.

a. Regulation of Stress Response and Metabolism genes in the adar RNA seq promotes and argument that it is similar to insulin signaling. Many different biological pathways in C. elegans have these characteristics. To show there is a specific link, there should a correlation between the genes expressed in both pathways and other data to show specificity. It is not uncommon to get overlap in RNA seq sets. There is nothing in this study to show that insulin signaling is the most similar in gene expression. 

b. Dod genes: I was concerned that the dod genes were cherry picked from the original data set and were not specific to insulin signaling. The author's reply was the genes were indeed selected from a based on the name (downstream of daf-16), not expression strength or pathway enrichment. Although these genes were names based on responsiveness to daf-16, they are regulated in response to around 300 different biological responses listed in the WormBase expression database, therefore there is no basis for using them as a sensor or proxy for insulin signaling. For example, dod-17 can be regulated by nhr-49 and the PMK-1 map kinase pathway. These genes are down regulated in daf-2 animals in the present study, but removing daf-2 does not change the adr-2 effects, so it is not possible to make a mechanistic conclusions.

2. I had commented in Figure 4 A that it is not accurate to say that there is no effect with ard-1 based on the lack of statistical significance. The data points show wide variability; therefore this appears to be an issue of lack of technical reproducibility, not biological variation. This is not sufficiently addressed.

3. My concern with the conclusions in the 4B was that the method states the qPCR is normalized to the negative control. Since there is nothing IP'd in the negative control, what is source of the RNA?

4. Regarding figure 5, building mechanistic models based on negative data is problematic, as it is difficult to test alternative hypotheses to establish sufficient rigor. Figure 5 shows that deletion of pqm-1 has no effect on survival in heavy metal stress. Expression of neural pqm-1 does limit survival, but overexpression can have many unintended effects. As I stated in the previous review, B lacks a critical control, the adr-1 without the mutations in the RNA binding domain.

5. I also agree with the other reviewer that calling the CoCl2 hypoxic stress is overly broad, as it is a heavy metal and induces heavy metal stress. Although the authors state there are overlaps of this stress with hypoxia, there are independent effects as well. Therefore using hypoxia in the title and throughout the text is not appropriate.

---

## [Editor Report · Decision Letter 3]

23 Aug 2023

Dear Heather,

Thank you for the submission of your revised Research Article "ADAR-mediated regulation of PQM-1 expression in neurons impacts gene expression throughout C. elegans and regulates survival from hypoxia" for publication in PLOS Biology. On behalf of my colleagues and the Academic Editor, Wendy Gilbert, I am pleased to say that we can accept your manuscript for publication, provided you address any remaining formatting and reporting issues. These will be detailed in an email you should receive within 2-3 business days from our colleagues in the journal operations team; no action is required from you until then. Please note that we will not be able to formally accept your manuscript and schedule it for publication until you have completed any requested changes.

PRESS

Best wishes,

Richard

Richard Hodge, PhD

rhodge@plos.org

PLOS
